# Quality Evaluation of Light- and Dark-Colored Hungarian Honeys, Focusing on Botanical Origin, Antioxidant Capacity and Mineral Content

**DOI:** 10.3390/molecules26092825

**Published:** 2021-05-10

**Authors:** Alexandra Bodó, Lilla Radványi, Tamás Kőszegi, Rita Csepregi, Dávid U. Nagy, Ágnes Farkas, Marianna Kocsis

**Affiliations:** 1Faculty of Sciences, Institute of Biology, University of Pécs, 7624 Pécs, Hungary; alexandrabodo88@gmail.com (A.B.); davenagy9@gmail.com (D.U.N.); 2Department of Pharmacognosy, Faculty of Pharmacy, University of Pécs, 7624 Pécs, Hungary; radvanyililla25@gmail.com (L.R.); agnes.farkas@aok.pte.hu (Á.F.); 3Department of Laboratory Medicine, Medical School, University of Pécs, 7624 Pécs, Hungary; koszegi.tamas@pte.hu (T.K.); ritacsepregi93@gmail.com (R.C.); 4János Szentágothai Research Center, 7624 Pécs, Hungary

**Keywords:** honey, melissopalynology, antioxidant activity, multielement analysis, PCA statistics

## Abstract

Melissopalynology, antioxidant capacity and mineral and toxic element contents were analyzed in eight types of Hungarian honeys. Based on color, two groups were distinguished: light honeys comprised acacia, amorpha, phacelia and linden honeys; while dark honeys included sunflower, chestnut, fennel and sage honeys, with 100 to 300 and 700 to 1500 mAU, respectively. The unifloral origin of each sample was supported using pollen analysis. The absorbance of honey correlated positively with antioxidant capacity determined by three different methods (TRC, DPPH, ORAC), and also with mineral content. The exception was the light amber linden honey with significantly higher K content and antiradical activity than other light honeys. The Mn, Zn and Fe contents were the highest in chestnut, sunflower and fennel honeys, respectively. The black meadow sage honey performed best regarding the content of other elements and antioxidant activity. The concentrations of several toxic elements were below the detection limit in the samples, indicating their good quality. The principal component analysis (PCA) revealed correlations between different antioxidant assays and minerals, and furthermore, confirmed the botanical authentication of the honeys based on the studied parameters. To our best knowledge, the present study is the first to provide a complex analysis of quality parameters of eight unifloral Hungarian honeys.

## 1. Introduction

Honey is a complex food, which has played an important role in human nutrition and medicine since ancient times. The essential constituents responsible for the dietary, antiviral, antifungal and antibacterial effects of the substance originate from single or multiple plant species. Consequently, the floral source is one of the most significant factors in determining the main quality parameters of honey [1]. The possible floral markers of honeys can be divided into pollen and phytochemicals [2]. For determining the pollen quality and quantity in the honey sample, the time-consuming melissopalynology method is the effective tool [3]. The chemical composition of honey depends on several factors, such as geographical origin, season and storage, so it is difficult to find reliable chemical markers. Therefore, the complex study of honey characters and their correlations are needed for establishing the proper authenticity of honey types.

There is a huge amount of studies about characterization of different honeys from different aspects, proving the importance of this natural food from plant origin. One of the most studied parameters of the honey is the antioxidant capacity, due to its richness in antioxidant compounds, which are synthetized by plants partly as secondary metabolites against reactive oxygen species [2,4]. Even in the last few years, several researchers have measured the total antioxidant parameters along with other properties of honey types from different countries, such as the characterization of Turkish honeys by Gül et al. [5], Polish honeys by Dzugan et al. [6], Goslinski et al. [7] and Halagarda et al. [8], Greek honeys by Stagos et al. [9], Serbian honeys by Sreckovic et al. [10], Romanian honeys by Pauliuc et al. [11] or Hungarian honeys by Bodó et al. [12]. The researchers used various methods for determining total antioxidant capacities (TAC), such as the 1,1-diphenyl-2-picrylhydrazyl (DPPH) assay for antiradical activity based on electron transfer and the oxygen radical absorbance capacity (ORAC) method based on hydrogen atom transfer [13]. Total reducing capacity (TRC) is also an effective tool for evaluation of TAC, generally called as a method for measurement of total polyphenol content (TPC). However, the Folin reagent used in this assay is nonspecific for polyphenols [14]. It has been widely established that the antioxidant properties of honeys primarily depend on their botanical origin. The correlation between honey color and antioxidant activity has also been proved [8,15,16,17]. To a lesser extent, environmental and climatic conditions, harvesting treatment and storage can also influence the antioxidant activity of honey [18].

Minerals are minor constituents of honey, but they play an important role in determining its quality [19]. Recent studies have been conducted also on mineral content of honeys: Altun et al. [20] analyzed Turkish honeys, Sager et al. [21] analyzed Austrian honeys, Conti et al., [22] analyzed Italian honeys and Czipa et al. [23] and Sajtos et al. [24] analyzed Hungarian honeys. The mineral composition of honey reflects not only the soil mineral content, but it also has strong botanical specificity [25,26]. According to Mohammed et al. [27], the mineral content of honeys can be used as a long-time marker of botanical origin and quality. The discriminating power of elements can help in classifying of unifloral honeys [28]. Correlation has also been established between the mineral composition and color of the honey, in so far as dark honeys contain higher amounts of certain minerals when compared to pale-colored ones. Moreover, the analysis of minerals and heavy metals in honey gives an indication of its geographical origin, contamination, and also an overall measure of honey purity [19]. Fewer studies were devoted to analyzing the correlation between antioxidant activity and mineral content. Indian and Turkish honeys were differentiated among others according to their macroelements and antioxidant capacities [29,30]. The effects of botanical specificity, geographical origin and harvest year on the antioxidant capacity and mineral content of Hungarian honeys were investigated in Bodó et al.’s [12] complex study. Perna et al. [31] found a significant effect of botanical origin on antioxidant activity, while other factors also influenced the mean metal content of honeys studied. They found correlations between the metal content and antioxidant activities of honeys as well.

The aim of this study was to give a complex analysis of Hungarian honeys representing eight honey types from light- to dark-colored repertoire, and furthermore, to reveal the correlations among their analyzed parameters. Chosen honeys included representatives from cultivated plants with high nectar production, to relatively rare specialty honeys, which are produced by few beekeepers in Hungary as well as worldwide. The comprehensive analysis included melissopalynology, antioxidant measurements with three different assays, a macro- and microelement profile, and furthermore, a toxic element analysis for their potential risk to human health. Although Hungary is one of the biggest honey producers in the EU with several uni- and multifloral honeys, there is little information on the comparative quality parameters of Hungarian honeys. The results contribute to the more extensive data of the honey samples studied and the correlations provide reliable characteristics of these natural products.

## 2. Results and Discussion

### 2.1. Pollen Study, Sensory Characteristics and Color of Honey Samples

The botanical origin of honey samples was established by microscopic pollen analysis and spectrophotometric color determination (Table 1, Figure 1), which is a combination of methods accepted for identifying different honey types [15,32]. In addition, further sensory characteristics, such as odor and consistency, were evaluated to confirm the floral origin of honey samples.

The results of the melissopalynological analysis revealed that each of the eight honey types were unifloral. With the exception of fennel and meadow sage honeys, the samples contained at least 45% of the characteristic pollen type, which is generally required to classify a honey as unifloral, if not specified differently. However, there are pollen types that are typically under-represented in honeys, which is often the case with honeys of Lamiaceae origin, e.g., in thyme honey the percentage of *Thymus* spp. pollen must only be at least 18% of the total pollen grains [33]. The sage honeys examined by Kenjerić et al. [34] contained 20–65% sage (*Salvia*) pollen, which was well above the required threshold value of at least 10% [35], and even complied with Croatian regulations of at least 20% *Salvia* pollen. In our honey samples, the percentage of sage (Lamiaceae) pollen grains was above 20%; thus, we considered meadow sage honeys as unifloral ones, even though they contained few honeydew elements, too. According to the classification [35], the 0.28 HDE/P (honey dew elements/absolute pollen count) value determined for sage honeys corresponds to a low proportion of honeydew elements. Similarly to Lamiaceae honeys, honey types of Apiaceae origin also tend to be under-represented. In fennel honeys investigated by Manzanares et al. [36], *Foeniculum* pollen was present in 2.6–45.7% (average: 12%). Since our fennel honey samples contained ca. 20% of Apiaceae pollen, we considered these honeys as unifloral, also supported by sensory characters. Similarly to sage honeys, fennel honeys contained few honeydew elements (HDE/P = 0.30).

Acacia (*Robinia*) and linden (*Tilia*) honeys also belong to the category with under-represented specific pollen. In the comprehensive study of Oddo et al. [37], the mean values for acacia and linden pollen percentages were 28% and 23%, with extreme values of 7–60% and 1–56%, respectively. Kuś et al. [15] found only 11–16% and 22–26% characteristic pollen grains in acacia and linden honeys, respectively. The percentage values of 45% *Robinia* pollen and 46% *Tilia* pollen in our acacia and linden honey samples, respectively, support the unifloral origin of these honeys. Kuś et al. [15] characterized the color of their linden honey samples as extra white, except for the sample with 26% *Tilia* pollen, which was reported as light amber, similarly to the color of linden honeys in our study. The consistency of Polish linden honey was described as solid, fine granulated, similarly to our linden honeys.

Oddo et al. [37] reported the percentage of *Helianthus* pollen in sunflower honeys to range from 12% to 92%. All of the 50 studied sunflower honey samples from Turkey contained 45–70% dominant pollen grains, proving their unifloral origin [38]. Similarly to our sunflower honey samples, they characterized the honey type as having a bright yellow color, fragrant smell with an aroma of pollen, creamy quality and fine texture, which crystallizes quickly. Besides these sensory characters, the 46% *Helianthus* pollen in our sunflower honeys supports the unifloral origin of these honeys.

From the eight honey types included in our study, phacelia and chestnut honeys can be characterized by over-represented pollen, with more than 60% *Phacelia* and at least 90% *Castanea* pollen, respectively [36,37]. These threshold values were fulfilled by our phacelia and chestnut honey samples, with mean values of 74% *Phacelia* and 91% *Castanea* pollen, respectively.

Color represents an important characteristic of honey, referring to its botanical origin and also its composition [39]. On the basis of the color, the studied honeys were divided into light- and dark-colored groups, exhibiting absorbance intensity from 100 to 300 mAU (acacia, amorpha, phacelia and linden honeys) and from 700 to 1500 mAU (sunflower, chestnut, fennel and meadow sage honeys), respectively. The reported color intensities were similar to the Romanian honeys, which were in the range of 210 mAU (acacia honey) to 1228 mAU (forest honey) [40]. Dżugan et al. [6] divided the studied honeys into light- and dark-colored honeys, exhibiting color intensity below or above 1 mAU, respectively (presumably meaning 1000 mAU, which would be comparable to all other similar studies). Net absorbances of Slovenian and Croatian honeys increased in an order similar to our results: acacia, linden and then chestnut [41,42]; however, their color values were lower than those of the same honey types in our study. Beretta et al. [43] reported a lower color parameter for acacia (25 mAU) and chestnut honeys (610 mAU), and Cimpoiu et al. [40] measured higher color intensity for acacia (210–295 mAU), but lower for sunflower honey (512–556 mAU) compared to our results. On the basis of a Pfund scale, honey can be classified as water white, extra white, white, extra light amber, light amber, amber and dark amber [44]. According to this classification, Sreckovic et al. [10] characterized the Serbian acacia honey as water white, while Hungarian acacia honey was slightly darker and classified as extra white [45]. The color of sunflower honey samples were extra light amber based on the Pfund scale [11].

Several researchers indicated that the color of honey is a reliable index of antioxidant activity and this sensory parameter is dependent on chemicals such as polyphenols and mineral content [18,19,46].

### 2.2. Total Antioxidant Capacities of Honeys

In the current study, three different TAC methods were used to determine the antioxidant behavior of the honey samples. The results varied in a wide range with significantly higher values in the case of dark honeys than in the case of light honeys (Table 2). Regarding the color, TRC and ORAC parameters, the dark honeys had much wider range of values than that of the light honeys. The colors distinguished all of the honeys from each other, while TRC, DPPH and ORAC of amorpha and phacelia did not differ significantly. Several researchers confirmed that acacia honey was the honey with the lowest antioxidant values [40,41,43]. In the dark group, the TRC of sunflower and chestnut honeys gave similar results, while ORAC did not distinguish the light-colored linden from the dark-colored chestnut. For the latter case, there are also some other exceptions, e.g., the relatively lighter arbutus honey and sourwood honey with especially high antioxidant power, which is similar to that of dark honeys [47]. The high antioxidant properties of chestnut honey were proved by several researchers [17,30,31]. In this study, the TAC results reached the maximum value in the case of meadow sage honey. Gośliński et al. [7] reported the highest TRC value for buckwheat and honeydew honeys with 200 mg GAE/100g. The observation that light-colored honeys possessed lower antioxidant activity compared to dark-colored ones [16] seems to also be valid for our study, except in the case of linden honey.

A high variation of TRC of different honeys was indicated by research conducted in different parts of the world. In this study, the lowest TRC value was provided by acacia, consistent with previous reports about Italian [43] and Slovenian [41] honeys. Our results are in accordance with those of the above-mentioned researchers, also in the case of the TRC of chestnut honey, but in the case of linden honey, their values were slightly lower. Flanjak et al. [42] measured somewhat lower values for these three types of honeys from Croatia, while Kus et al. [15] obtained higher TRC parameters for acacia (142.8 ± 16 mg·GAE·kg^−1^) and linden (192.5 ± 17.8 mg·GAE·kg^−1^) than our results. Furthermore, Gül et al. [5] measured even higher reactivities for both honey types (51.91 mg·GAE·kg^−1^, 268.81 mg·GAE·100 g^−1^, respectively) and also for chestnut honey (327.60 mg·GAE·100 g^−1^). However, the same tendency could be observed in each study, establishing the following order of the Folin reactivity: acacia ˂ linden ˂ chestnut honey. The TRC values of sunflower honey in this study were in line with those reported by Pauliuc et al. [11]. Sari et al. [38] summarized the parameters of 50 sunflower honeys in Turkey, providing a broad range of TRC for this honey type (6.9–23.2 mg·GAE·100 g^−1^). Our samples approached the upper limit reported by them. Similarly to acacia, for the linden and chestnut honeys, much higher values were calculated by Gül et al. [5] for these honey types in Turkey (77.64 ± 0.86 mg·GAE·100 g^−1^). The darkest colored samples in our study, i.e., fennel, and particularly, meadow sage honeys, had high TRC values. Values of the latter fell above the range of 0.55 to 0.92 mg·GAE·g^−1^, measured for multifloral honeys from Greece [9].

The DPPH assay was used to determine the free radical-scavenging activity of the honey samples. In this case, the lower the IC_50_ value, the higher the antioxidant activity. In this study, the highest DPPH radical-scavenging activity (Table 2) was identified for meadow sage honey and the lowest for acacia honey. Values varied significantly (*p* < 0.01) depending on the botanical origin of the honey samples, except for the light-colored acacia, amorpha and phacelia honeys, whose IC_50_ values did not differ from each other. The assay has been frequently used to characterize the antioxidant activity of honeys, e.g., Polish honeys [6,7,15], Czech honeys [48], Romanian honeys [11], Indian honeys [29] or Lithuanian honeys [49]. Beretta et al. [43] and Bertoncelj et al. [41] reported similar values for acacia honey compared to our results, while Flanjak et al. [42] measured much lower activity (125.48 mg·mL^−1^). Regarding linden honey in this study, its activity was in the upper range measured by Bertoncelj (28.8 ± 5.4 mg·mL^−1^) and in the lower range measured by Flanjak (42.77 ± 10.32 mg·mL^−1^). The activity of chestnut honey in this study was lower than those obtained by the above-mentioned researchers. The multifloral honeys from Greece exhibited a broad range of DPPH activity from 7.5 to 109.0 mg·mL^−1^ [9]. The meadow sage honey in this study displayed higher antioxidant activity even compared to these Greek honeys.

Compared to other assays used for characterizing the antioxidant capacity of honeys, the ORAC assay has been used less frequently. In line with the results of TRC and DPPH, the ORAC activity of dark honeys was significantly higher than that of the light honeys (Table 2). This assay, however, separated linden honey from the light-colored group with a similarly high ORAC value as is seen in chestnut honey. In agreement with our results, one of the lowest ORAC activity values was found in acacia honeys [13,50]. The ORAC value of chestnut honey was about four-times higher than that of acacia honey [43].

### 2.3. Multielement Analysis of Honeys

The macro- and microelement contents determined in our honey samples are summarized in Figure 2 and Table 3 and Table 4. Detailed mineral parameters of our samples from the western part of Hungary were compared primarily with the results of honey samples from others in the southern and eastern part of the country [23,24,51].

Regarding the macroelement content (Figure 2a, Table 3), the dark-colored meadow sage honey was found to be particularly rich in minerals, while the light-colored phacelia honey was found to be poor in these elements, with average values of 3497 mg·kg^−1^ and 222 mg·kg^−1^, respectively. Within the pale honey group, excluding linden honey, the total macroelement content was below 300 mg·kg^−1^, while in other honeys, it was above 1000 mg·kg^−1^. Expectedly, K was found to be the most abundant mineral in all studied honeys. Linden honey had significantly higher K content, and consequently, total macroelement content (1429 mg·kg^−1^), than the other light-colored honeys, even higher than the darker sunflower honey (1034 mg·kg^−1^). The mean K level in acacia and linden honey was comparable to that reported for these honey types in Croatia [52]. Relatively high amounts of K and Ca in linden and chestnut honey were found also in samples from different parts of Hungary [23,24]. These results are in accordance with our observation regarding the K content in the following order: phacelia ˂ acacia ˂ sunflower ˂ linden ˂ chestnut. The significantly higher content of K and Ca in chestnut honey compared to those in acacia honey was confirmed previously by Kaygusuz et al. [30]. Comparing Ca to P and S to Mg contents within the same honey provided differences (*p* ˂ 0.05) among honey types as follows: linden, sunflower and chestnut honeys had higher Ca content than P, while the pale acacia, amorpha, phacelia and the dark fennel and meadow sage honeys showed a reverse relation. The absolute values obtained for the above-mentioned five honey types were not uniform among the studied Hungarian honeys [23,51]; however, the tendency regarding their average amounts of Ca and P showed considerable correlation. S and Mg also had different relations in honey types. S content was higher than Mg in the light-colored acacia, amorpha and phacelia honeys, while in the dark fennel and meadow sage, Mg preceded the S. Linden, sunflower and chestnut honeys contained similar amounts from both elements. Similarly to our results, the S content of acacia and phacelia was significantly higher than their Mg content [23]. The range of Mg content was similar to that of honeys reported from Bulgaria, France, Italy and Poland; only that of fennel and meadow sage honey was above the upper limit [19]. Among the macroelements, the amount of Na was the lowest, except in acacia and amorpha honey, and in the latter, Na proved to be the third main macroelement after K and Ca. Nayik et al. [29] measured higher amount of Na than P content in Indian acacia honey. A high Na level also characterized the avocado honey, in which Na was the second most abundant mineral [53]. The range of Na content differed significantly between continents, e.g., honeys from Europe (Bulgaria, Italy, Poland) contained a low amount of Na, while much higher Na levels have been detected in India or Malaysia [19]. Compared to our results, the variety of macromineral content of Italian multifloral honeys decreased in a different order (K ˃ S ˃ Ca ˃ P ˃ Na ˃ Mg) [22].

The amounts of microelements determined in our honey samples are summarized in Figure 2b and Table 4. All of them were present in the dark honeys, while Cu, Fe and Mn were under the detection limit in some of the light-colored ones. Consequently, the dark-colored group contained a significantly higher amount of microelements than the light group. Chestnut honey was the richest and amorpha honey the poorest in microelement content with the average amount of 17.3 mg·kg^−1^ and 3.0 mg·kg^−1^, respectively. Based on the microelements, phacelia was distinguishable from other light-colored honeys due to its significantly higher B and Zn content, and chestnut honey stood out from dark-colored honeys due to its extremely high Mn content. Among the studied microelements, B content showed the highest amount in all honey types, except chestnut honey. However, Amtmann et al. [51] measured lower B content in acacia, sunflower, chestnut and even phacelia honey (0.98 mg·kg^−1^) compared to our results, while that of linden honey was similar in the two studies. In contrast, Sajtos et al. [24] measured higher B content in the above-mentioned honeys, compared to our and to Amtmann et al.’s [51] results. Similarly to our results, the Cu content of acacia and phacelia was very low or under the detection limit in the study of Czipa et al. [23]. They reported a similar average amount of Cu in sunflower honey, while finding a higher Cu content in linden honey (0.320 ± 0.073 mg·kg^−1^). Furthermore, the mean Fe content of acacia and sunflower honeys was similar to our results, but that of phacelia honey was lower (0.225 mg·kg^−1^) and that of linden honey was higher (0.612 mg·kg^−1^) than in our study. The highest Fe content was measured in fennel honey, but a much higher upper limit of Fe content had been detected in honeys from France (86.76 mg·kg^−1^) and Malaysia (233 mg kg^1^) [19]. Among the microelements, chestnut honey was highlighted by a particularly high Mn content, in accordance with the results of Sajtos et al. [24], who measured an even higher level of Mn in this honey type (11.9 mg·kg^−1^). The high Mn content of chestnut honey was confirmed by several authors, being similarly high as that of fir honey, but slightly lower than that of oak honey [25,30]. The highest Zn level was found in our sunflower honey samples, supporting the observations of Sajtos et al. [24], who compared the Zn content of sunflower honey to that of acacia, linden, phacelia and chestnut honeys. Similarly to our results, Bogdanov et al. [25] showed higher Cu, Fe and Zn content in chestnut honey than in acacia. Regarding the linden honey in the above-mentioned study, it showed similar Cu and Fe content, and a higher Zn content than chestnut honey, while in our study, linden honey contained a lower amount of each microelement than chestnut honey. Compared to our results, higher Cu and Fe levels have been reported for acacia, linden and chestnut honeys in Croatia, while the Zn content of chestnut honey was higher in our samples [52].

The third mineral group was composed of nine trace elements: Al, As, Cd, Co, Cr, Mo, Ni, Pb and V. One out of the three samples of linden, sunflower and chestnut honeys contained Al (1.07, 1.04 and 1.76 mg·kg^−1^, respectively). Chestnut, fennel and meadow sage honeys displayed quantifiable amounts of Cd (0.12 ± 0.028, 0.22 ± 0.01 and 0.40 ± 0.005 mg·kg^−1^, respectively), while Ni was measured in one sample of chestnut honey (0.15 mg·kg^−1^). The others were below the detection limit (< 0.1 mg·kg^−1^), which indicates that the nectar sources of the honeys in this study were not contaminated or only to a very minor degree. High concentration of heavy metals, such as Co, Cr, As, Cd, Hg and Pb in honey can be toxic for humans, causing metabolic abnormalities, respiratory problems or even damage of the kidneys and of the nervous system [54,55]. Solayman et al. [19] summarized the results of several elements measured in honeys from six continents. The upper limit of Al in honeys from India and Turkey was very high (16.12, 13.68 mg·kg^−1^, respectively), it was lower in some European countries (France 9.72 mg·kg^−1^ and Spain 8.31 mg·kg^−1^), and similar to the value of our chestnut honey in Bulgaria (1.58 mg·kg^−1^). The Cd content of some honeys in Croatia and Romania reached the amount of 4000 µg·kg^−1^ and 1600 µg·kg^−1^, respectively, while it was not detected in Polish honeys.

The amount of the elements can be affected by climatic, geologic and environmental features according to Pisani et al. [56] and Islam et al. [57]; however, the minor components in honeys cannot be considered as a reliable bio monitor of environmental pollution [22]. Mohammed et al. [27] showed evidence that minerals in some honeys were not in strict relation with the soil mineral content. Bogdanov et al. [25], Nayik et al. [29] and Mohammed et al. [27] concluded that the mineral composition in honey has been primarily attributed to the botanical origin rather than the geographical and environmental exposition of nectar sources.

### 2.4. Correlation Analysis

The data matrix of color, antioxidant values and multielement contents was analyzed by Pearson’s correlation and PCA methods to obtain more information (Table 5, Figure 3). The correlation matrix showed a significant relationship between all the variables, except in the cases of Mn and Zn. However, differences between correlation coefficients provided useful information on the measured parameters of honeys.

High linear correlation was obtained between color and TAC values, and among the TAC assays. Similarly to our results, many studies observed that darker honeys had a higher TAC, while lighter honeys were characterized by relatively low TAC values [43,58]. The DPPH assay was the strongest predicting factor regarding the absorbance (color) of the honey samples, supported also by Dzungan et al. [6], Flanjak et al. [42] and Gorjanovich et al. [13]. Although different TAC assays may reflect different antioxidants, the used methods gave a positive linear correlation in this study, consistent with the results of other authors [10,15,17,29]. In our case, the strongest correlation was found between DPPH and ORAC, while the lowest correlation was found between DPPH and TRC, and the latter was also proved by Stagos et al. [9] and Perna et al. [31].

All of the studied macro- and microelements were in good correlation with color, except Mn (r = 0.257; *p* ˃ 0.05). The contribution of minerals to determine honey color has already been supported [59]. Perna et al. [31] proved that color intensity is positively correlated with metal content in honey; furthermore, with an increase in metal content, there is an increase in antioxidant capacities of honeys. Our study confirms the possible existence of the link between the following antioxidant assays and macroelements: TRC with Mg, P, S contents, DPPH with Ca content and ORAC with K content of honeys. The microelements, B, Cu and Fe significantly correlated with the antioxidant capacities. B showed the highest correlation with DPPH, while Cu showed the highest correlation with TRC and ORAC. Zn correlated only with DPPH (r = 0.717; *p* ˂ 0.001), while Mn did not show a correlation with any of the antioxidant capacities (TRC: r = 0.036, DPPH: r = 0.345, ORAC: r = 0.313; *p* ˃ 0.05).

The above-described correlations could be clearly interpreted in light of the PCA. The score plot (data not shown) indicates that the first two principal components account for 98% of the total variance. The first principal component, PC1, includes most of the information (92% of the total variance), while the second principal component, PC2, explains 5% of the total variance. The original data set was renormalized by an autoscaling transformation since the values for the various parameters are expressed in different units. Light-colored honeys with lower antioxidant activities and mineral content were located on the negative PC1. Consequently, dark honeys with high parameters were located on the positive PC1 values of the plot. A similar arrangement was observed on the PC2 coordinate, except in the case of linden honey with a positive PC2 value. The light-colored acacia and amorpha were located close to each other. Most of their analyzed parameters showed a low variability, suggesting the close relation of their plant origin (Fabaceae). The K content and ORAC activity were useful in clustering linden honey separately. The samples selected for the dark group differentiated significantly from each other; therefore, it was easy to distinguish them based on the antioxidant and multielement values. Zn and Mn values played a key role in the clustering of sunflower and chestnut honeys, respectively. Pauliuc et al. [11] applied a partly successfully PCA of physicochemical parameters to analyze and identify honey types with similar characteristics from different regions in Romania. Dzugan et al. [6] found that various antioxidant activities could refer to the botanical origin of the honeys. Nayik et al. [29] concluded that macroelements and antioxidant properties had a higher discriminating power in the case of three Indian honey types than sugar parameters. Values of vitamin B2, antioxidant activities and mineral contents successfully clustered 20 monofloral and honeydew honeys from Turkey [30].

However, in this study, PCA was applied not only to identify honey types, but also to interpret the relationships of the studied parameters, namely, the antioxidant activities, macroelement contents and botanical origin of honeys.

## 3. Materials and Methods

### 3.1. Samples

The honey samples were purchased from the producers. Honeys were harvested in the Southwest Transdanubium area in Hungary in 2018. They were stored at room temperature (20–21 °C) in the dark for a maximum of three weeks. For each honey type (Table 1), measurements were carried out on 3 parallel samples; altogether, 24 honey samples were analyzed.

### 3.2. Melissopalynological Analysis

Melissopalynological analysis was carried out according to Von Der Ohe et al. [60]. The pollen was examined under a microscope using 400× magnification on a Motic microscope (Motic BA310, Electromed Kft., Budapest, Hungary). To determine the botanical origin, at least 500 pollen grains were counted. The percentage frequency of the dominant pollen types was calculated in all honey samples.

### 3.3. Determination of Color Intensity (ABS_450_)

The color intensity of honey samples was determined at the wavelengths 450 nm and 720 nm, according to Beretta et al. [43]. The absorbance was measured using a Shimadzu UV-1800 spectrophotometer (Shimadzu Schweiz GmbH, Reinach, Switzerland), and the results were expressed as a milli-absorbance unit (mAU).

### 3.4. Total Reducing Capacity (TRC)

The TRC was determined using the Folin–Ciocalteau method as reported by Singleton et al. [61]. The results were expressed as mg gallic acid equivalents (GAE) kg^−1^ of honey. A GA (Fluka Chemie AG, Buchs, Switzerland) (50–200 µg·mL^−1^) was used to obtain the calibration curve.

### 3.5. Antiradical Power (DPPH)

The antiradical activity of honey samples was measured using the method of Beretta et al. [43] and Bertoncelj et al. [41] with some modifications. For the assay, 4 mg of DPPH in 50 mL of 96% absolute ethanol (200 µmol·L^−1^) was prepared and kept in the fridge. Trolox standards were prepared in 100 mM acetate buffer at pH 5.5 (100 mM acetic acid and 100 mM sodium acetate trihydrate), in the concentration range of 0–180 µmol·L^−1^. The assay was adapted to a plate reader (Perkin Elmer EnSpire Multimode reader, Waltham, MA, USA) using standard 96-well plates (Sarstedt AG & Co. KG, Nümbrecht, Germany). Into each well, 50 μL of the blank/standard/sample, 95 μL of DPPH solution and 50 μL of acetate buffer solution were added. The mixture was shaken and the absorbance changes were measured at 517 nm after 60 min of incubation in the dark at room temperature (25 °C). The radical-scavenging activity was expressed as IC_50_ (the concentration of the honey sample (mg·mL^−1^) needed to scavenge 50% of DPPH), calculated using a linear regression analysis.

### 3.6. Oxygen Radical Absorbance Capacity (ORAC)

The ORAC assay was based on the procedure previously described by Kőszegi et al. [62] and Patay et al. [63] without modifications. Briefly, a fluorescein working solution (400 nmol·L^−1^) and the 2,2′-azobis(2-amidinopropane) dihydrochloride (AAPH) oxidant (400 mmol·L^−1^) dissolved in 75 mmol·L^−1^ potassium phosphate buffer (mixture of KH_2_PO_4_ and K_2_HPO_4_) at pH 7.5 were prepared freshly before the measurements. Trolox standards were prepared in the potassium phosphate buffer (0–160 µmol·L^−1^). Into each well, 25 µL of the blank/standard/sample and 150 µL of fluorescein solution were added in optical plates (Perkin Elmer) and the mixture was incubated at 37 °C for 30 min in the dark. Next, 25 µL AAPH solution/well was injected by the automated injector of a Biotek Synergy HT plate reader (BioTek Instruments, Winooski, VT, USA) and warmed up to 37 °C. The fluorescence intensities were monitored for 80 min (490/520 nm wavelengths) at 2 min intervals. The area under each curve (AUC) was obtained using the software of the reader providing the total sum of the individual digital data of the corresponding fluorescence signals. The antioxidant capacity values were expressed as µmol Trolox equivalent (TE) g^−1^ honey.

### 3.7. Inductively Coupled Plasma Atomic Emission Spectrometry (ICP-AES)

ICP-AES measurements of 20 elements were done using an ICPE-9000 instrument (Shimadzu, Kyoto, Japan) at the following operating parameters: radio frequency power, 1.20 kW; plasma gas, 10.0 L·min^−1^; auxiliary gas, 0.60 L·min^−1^; carrier gas, 0.70 L·min^−1^; and view direction, axial. Prior to the elemental analysis, honey samples were pretreated using a Multiwave 3000 (Anton Paar GmbH, Graz, Austria) microwave system, in which 1 g of each honey sample was treated in three steps: 300 W for 5 min, 1000 W for 5 min and 1400 W for 20 min. The instrument was calibrated using inorganic reference standards for a single element (BDH Prolabo Chemicals, VWR International Kft., Debrecen, Hungary). Quality control was assured using a high-purity multielement standard solution containing 25 elements (HPS, RK Tech Kft., Budapest, Hungary). A recovery test was done by spiking rape honey with 20 ppm of the ICP multielement standard mixture. Recoveries for the 20 elements ranged from 93.8% to 111.5%. All analyses were carried out in triplicate. Detection limits (LOD) were as follows: 15.0 mg·kg^−1^ for K, 10.0 mg·kg^−1^ for Ca, 5.0 mg·kg^−1^ for S and Na, 2.0 mg·kg^−1^ for Mg, 1.5 mg·kg^−1^ for P, 1.0 mg·kg^−1^ for B and Al, 0.5 mg·kg^−1^ for Fe and Pb and 0.1 mg·kg^−1^ for As, Cd, Co, Cr, Cu, Mn, Mo, Ni, V and Zn.

### 3.8. Statistical Analysis

All measurements were done on three biological replicates of eight honey types. Statistical analyses were carried out using Excel^®^ (Microsoft Corp., Redmond, WA, USA) and the paleontological statistics software package (PAST) version 3.11 [64] at a 5% significance level (*p* < 0.05), after normality checking with the Shapiro–Wilk test. For the correlation matrix, the 1% (*p* < 0.01) and 0.1% (*p* < 0.001) significance levels were used to indicate the greater significance of the differences. Data were expressed as means ± standard deviations (SD). Pairwise comparisons were performed with Student’s *t*-tests. Interactions between the measured parameters were investigated with Pearsons’ rank correlation using PAST. To describe relatedness among honey types, we performed a centered and standardized principal component analysis (PCA) with all measured parameters using the ggfortify 0.4.8. package [65] in R, version 3.5.3 (R Development Core Team 2019, R: A language and environment for statistical computing. R Foundation for Statistical Computing, Vienna, Austria. URL https://www.R-project.org/, accessed on 7 April 2021). Distances among object points (honey types) were calculated with Euclidean distances.

## 4. Conclusions

Our study on honey samples ranging from pale to black color verified and supported the complex relationship between the botanical origin, antioxidant property and mineral content of honeys. The light-colored linden honey, whose activity was comparable to the dark-colored chestnut honey, was an exception to the general observation that dark honeys presented better parameters as compared to light honeys. The antioxidant properties and mineral content of the black-colored fennel and meadow sage honeys were characterized for the first time in this study. The obtained results showed that minerals had high discriminating power to characterize honeys, and there was clear evidence for their role in the bioactive function. The present study indicated a strict correlation between the ORAC antioxidant assay and the most important macroelement, potassium. This mineral was the most variable in regard to botanical origin, proving its useful discrimination/identification properties. The correlation matrix and application of multivariate analysis suggested further possible relationships between the antioxidant and mineral parameters and put them forward as a useful tool to characterize different types of honey. We can conclude that the revealed markers gave the possibility to assign a characteristic fingerprint to a given honey, which refers to its botanical origin, quality and identity.

## Figures and Tables

**Figure 1 molecules-26-02825-f001:**
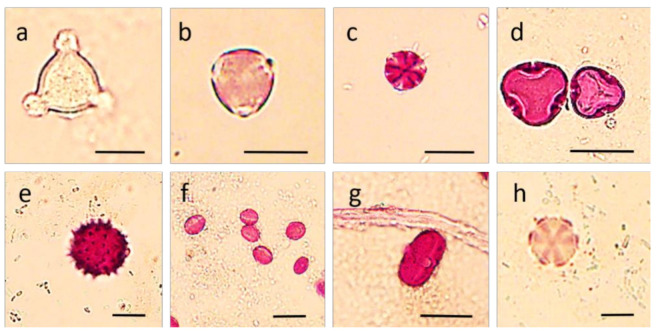
Characteristic pollen grains observed in the honey samples. (**a**) Acacia—*Robinia pseudoacacia*, (**b**) Amorpha—*Amorpha fruticosa*, (**c**) Phacelia—*Phacelia tanacetifolia*, (**d**) Linden—*Tilia* spp., (**e**) Sunflower—*Helianthus annuus*, (**f**) Chestnut—*Castanea sativa*, (**g**) Fennel (*Foeniculus vulgaris*)—Apiaceae pollen, (**h**) Meadow sage (*Salvia pratensis*)—Lamiaceae pollen. Scale bar = 25 µm.

**Figure 2 molecules-26-02825-f002:**
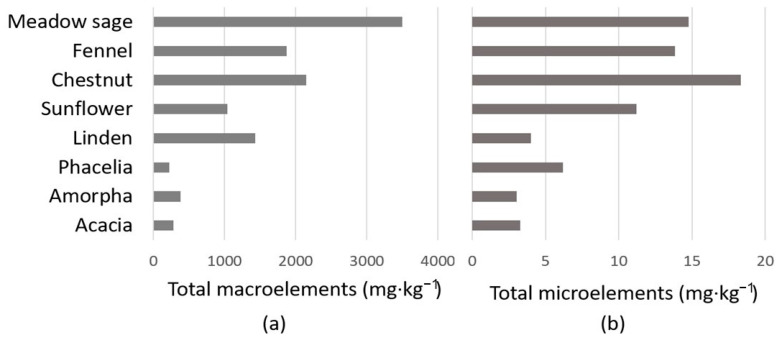
Reported average concentrations of (**a**) macroelements and (**b**) microelements in the studied honeys.

**Figure 3 molecules-26-02825-f003:**
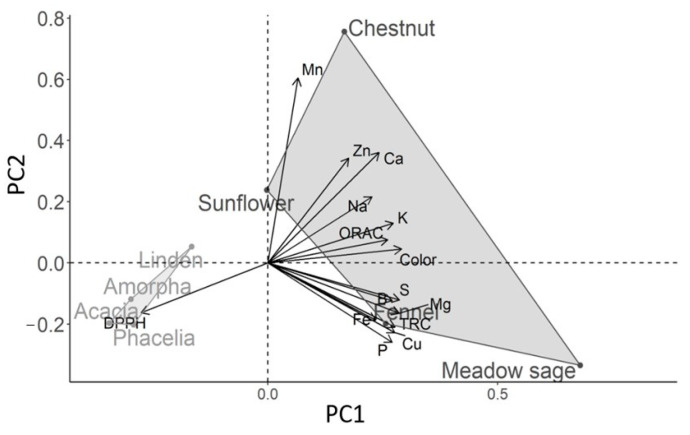
Scatter plot generated from a correlation matrix PCA indicating the direction of each variable and the position of each honey sample on the PC1–PC2 independent coordinates.

**Table 1 molecules-26-02825-t001:** Identification and sensory characteristics of analyzed honey samples.

Nr.	Honey Type Plant Name	Dominant Pollen (%)	Sensory Characteristics (Color, Odor and Consistency)	ABS_450_ (mAU)
1	Acacia *Robinia pseudoacacia*	*Robinia pseudoacacia* (45.27%)	Pale, yellowish green, weak odor, liquid, viscous	136 ± 4
2	Amorpha *Amorpha fruticosa*	*Amorpha fruticosa* (74.77%)	Pale yellow, weak odor, liquid, viscous	191 ± 5
3	Phacelia *Phacelia tanacetifolia*	*Phacelia tanacetifolia* (74.07%)	Light beige, moderately intense odor, fine granulated, semisolid	247 ± 11
4	Linden *Tilia* spp.	*Tilia* spp. (45.89%)	Light amber, strong odor, fine granulated, semisolid	285 ± 8
5	Sunflower *Helianthus annuus*	*Helianthus annuus* (47.44%)	Bright yellow, moderately intense odor, coarsely granulated, solid	719 ± 5
6	Chestnut *Castanea sativa*	*Castanea sativa* (90.81%)	Dark amber with reddish tone, strong odor, liquid, viscous	920 ± 10
7	Fennel *Foeniculum vulgare*	Asteraceae (21.97%) Apiaceae (19.23%)	Almost black, strong, caramel odor, liquid, viscous	1087 ± 33
8	Meadow sage *Salvia pratensis*	Lamiaceae (24.63%)	Black, strong, sage-like odor, liquid, viscous	1459 ± 52

Each code number in the first column represents three biological replicates (*n* = 3) of honey samples.

**Table 2 molecules-26-02825-t002:** Total antioxidant capacities of selected honey samples.

Nr.	Honey Types	TRC (mg·GAE·kg^−1^)	DPPH (IC_50_mg·mL^−1^)	ORAC (µmol·TE·g^−1^)
1	Acacia	60.08 ± 6.24 ^a^	61.76 ± 2.85 ^a^	19.81 ± 1.72 ^a^
2	Amorpha	80.81 ± 15.81 ^b^	55.48 ± 1.86 ^a^	14.78 ± 1.16 ^b^
3	Phacelia	91.67 ± 19.03 ^b^	55.78 ± 1.95 ^a^	13.79 ± 0.58 ^b^
4	Linden	119.14 ± 13.80^c^	35.86 ± 0.62 ^b^	71.68 ± 5.43 ^c^
5	Sunflower	230.25 ± 8.35 ^d^	26.62 ± 0.49 ^c^	34.32 ± 3.57 ^d^
6	Chestnut	232.82 ± 24.97 ^d^	17.37 ± 0.57 ^d^	75.20 ± 4.71 ^c^
7	Fennel	468.00 ± 73.16 ^e^	12.28 ± 0.25 ^e^	61.33 ± 5.83 ^e^
8	Meadow sage	1116.15 ± 83.84 ^f^	5.47 ± 0.02 ^f^	114.89 ± 10.43 ^f^
**Total**				
Light-colored honeys (nr. 1–4)		87.92 ± 25.67 ^a^	52.22 ± 10.35 ^a^	30.84 ± 25.01 ^a^
Dark-colored honeys (nr. 5–8)		511.81 ± 371.2 ^b^	15.43 ± 8.07 ^b^	71.54 ± 29.77 ^b^

TRC—total reducing capacity; DPPH—antiradical power; ORAC—oxygen radical absorbance capacity; Data are means ± standard deviations of three independent determinations (*n* = 3). Means in the same column with different superscripted letters are significantly different according to Student’s *t*-test (*p* < 0.01).

**Table 3 molecules-26-02825-t003:** Macroelement content of the studied honey samples.

Nr.	Honey Types	K (mg·kg^−1^)	Ca (mg·kg^−1^)	P (mg·kg^−1^)	S (mg·kg^−1^)	Mg (mg·kg^−1^)	Na (mg·kg^−1^)
1.	Acacia	226.56 ± 17.42 ^a^	12.39 ± 1.44 ^a^	24.92 ± 1.62 ^a^	7.07 ± 0.35 ^a^	5.24 ± 0.28 ^a^	5.99 ± 0.16 ^a^
2.	Amorpha	282.47 ± 19.22 ^b^	16.61 ± 2.50 ^a,b^	42.25 ± 3.65 ^b^	13.11 ± 1.44 ^b^	8.38 ± 0.29 ^b^	14.25 ± 2.05 ^b^
3.	Phacelia	145.62 ± 4.23 ^c^	19.88 ± 3.11 ^b^	33.68 ± 2.46 ^c^	13.04 ± 0.88 ^b^	6.04 ± 0.33 ^c^	3.56 ± 3.09 ^a^
4.	Linden	1278.08 ± 18.97 ^d^	67.85 ± 8.01 ^c^	41.52 ± 4.46 ^b,c^	15.89 ± 4.46 ^c^	16.51 ± 0.30 ^d^	9.29 ± 1.03 ^c^
5.	Sunflower	758.95 ± 18.69 ^e^	126.37 ± 14.93 ^d,e^	76.25 ± 8.22 ^d^	26.53 ± 8.22 ^d^	33.26 ± 1.28 ^e^	13.23 ± 1.70 ^b^
6.	Chestnut	1815.79 ± 20.69 ^f^	153.01 ± 12.60 ^d^	79.04 ± 5.41 ^d^	35.55 ± 5.41 ^d,e^	45.38 ± 17.32 ^e^	20.94 ± 0.80 ^d^
7.	Fennel	1373.99 ± 41.36 ^g^	103.46 ± 14.64 ^e^	251.85 ± 28.47 ^e^	44.91 ± 28.47 ^e^	86.82 ± 3.29 ^f^	9.14 ± 0.31 ^c^
8.	Meadow sage	2523.02 ± 28.45 ^h^	135.60 ± 21.16 ^d^	549.66 ± 54.64 ^f^	96.68 ± 54.64 ^f^	167.12 ± 2.08 ^g^	25.08 ± 2.76 ^e^

Data are means ± standard deviations of three independent measurements (*n* = 3). Means in the same column with different letters are significantly different according to Student’s *t*-test (*p* < 0.05).

**Table 4 molecules-26-02825-t004:** Microelement content of honey samples.

Nr.	Honey Types	B (mg·kg^−1^)	Cu (mg·kg^−1^)	Fe (mg·kg^−1^)	Mn (mg·kg^−1^)	Zn (mg·kg^−1^)
1.	Acacia	2.99 ± 0.21^a^	<0.10	<0.05	0.12 ± 0.02 ^a^	0.15 ± 0.08 ^a^
2.	Amorpha	2.58 ± 0.41^a^	<0.10	<0.05	0.13 ± 0.01 ^a^	0.31 ± 0.04 ^a^
3.	Phacelia	4.10 ± 0.52 ^b^	<0.10	0.91 ± 0.54 ^a^	<0.10	1.17 ± 0.25 ^b^
4.	Linden	2.70 ± 0.09 ^a^	0.12 ± 0.02 ^a^	<0.05	1.01 ± 0.03 ^b^	0.15 ± 0.08 ^a^
5.	Sunflower	4.90 ± 0.61 ^b,c^	0.23 ± 0.02 ^b^	0.75 ± 0.09 ^a^	0.45 ± 0.01^c^	4.87 ± 0.15 ^c^
6.	Chestnut	4.51 ± 0.44 ^b^	0.34 ± 0.20 ^a,b^	1.16 ± 0.85 ^a,b^	8.45 ± 2.81^d^	3.88 ± 1.23 ^c,d^
7.	Fennel	6.46 ± 0.97 ^c^	0.82 ± 0.03 ^c^	3.07 ± 0.47 ^c^	0.27 ± 0.03 ^e^	3.21 ± 0.18 ^d^
8.	Meadow sage	7.56 ± 0.74 ^c^	1.67 ± 0.03 ^d^	2.35 ± 0.26 ^b,c^	0.56 ± 0.01^f^	2.63 ± 0.20 ^e^

Data are means ± standard deviations of three independent measurements (*n* = 3). Means in the same column with different letters are significantly different according to Student’s *t*-test (*p* < 0.05).

**Table 5 molecules-26-02825-t005:** Correlation matrix (Pearson’s correlation coefficients) of color, antioxidant and macroelement parameters in Hungarian honeys.

Variable	Color	TRC	DPPH	ORAC
TRC	0.886 **			
DPPH	0.947 **	0.763 **		
ORAC	0.825 **	0.817 **	0.865 **	
K	0.879 **	0.816 **	0.908 **	0.983 **
Ca	0.845 **	0.607 **	0.914 **	0.754 **
P	0.860 **	0.968 **	0.731 **	0.783 **
S	0.920 **	0.966 **	0.817 **	0.844 **
Mg	0.929 **	0.980 **	0.825 **	0.843 **
Na	0.713 **	0.679 **	0.682 **	0.716 **
B	0.908 **	0.836 **	0.801 **	0.652 **
Cu	0.900 **	0.979 **	0.789 **	0.824 **
Fe	0.841 **	0.715 **	0.765 **	0.576 *

TRC—total reducing capacity; DPPH—antiradical power; ORAC—oxygen radical absorbance capacity; Significant at * *p* < 0.01, ** *p* < 0.001.

## Data Availability

The data presented in this study are available on request from the corresponding author.

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
