# Peer review of "Quality Evaluation of Light- and Dark-Colored Hungarian Honeys, Focusing on Botanical Origin, Antioxidant Capacity and Mineral Content"

_molecules, 2021, doi:10.3390/molecules26092825_

Round 1

Reviewer 1 Report

Work conducted in this study does not possess an innovative approach, since a lot of manuscripts are published with a similar concept. However, from the point of evaluation of quality and characterization of Hungarian honeys, I found that it can be scientifically important.

My recommendation is to include basic physicochemical analysis such as moisture content, pH, free acidity, HMF content, and electrical conductivity.

Author Response

Response to the comments from Reviewer 1

Thank you for the review and for your kind suggestions of our manuscript.

Work conducted in this study does not possess an innovative approach, since a lot of manuscripts are published with a similar concept. However, from the point of evaluation of quality and characterization of Hungarian honeys, I found that it can be scientifically important.

My recommendation is to include basic physicochemical analysis such as moisture content, pH, free acidity, HMF content, and electrical conductivity.

We agree, that measurements of some more basic parameters would be useful in characterization of the honey samples.

Unfortunately, we did not analyse the above mentioned physicochemical characters of the honeys during the experimental time period, so we think, it could be misleading to measure these from new, recent honey samples because it is not possible to have the very same honeys that are 100% comparable with those that have been characterized in the manuscript.

Of course, we are grateful for your comment, and we will complete the spectrum of the measured parameters with these features in the future. Also, the protein spectrum of the various honeys can be of interest performed by 1D SDS PAGE.

Reviewer 2 Report

The manuscript by Kocsis and colleagues characterized  eight selected Hungarian honeys based on their botanical origin, mineral content and the in vitro radical scavenging activity. The motivation behind the study is to generate a comprehensive picture of the characteristics of honeys produced in Hungary, one of the major honey producers in Europe. As expected from previous studies, the darkness of honey samples generally correlated with many other measurements, such as the results from the TRC, DPPH, and ORAC assays, as well as the content of many minerals (e.g., K, Ca, P, S, and Mg). The introduction is well-written and covers key literature in the field. The methodology is technically sound and the results are logically presented and explained. Despite some degree of speculation, the statements in the discussion and conclusions are supported by the experimental data and literature. Although it would be more interesting to also have performed antioxidant measurements using cell-based assays, the generated dataset tells a complete story and the manuscript reached the objective it was set out to achieve. Noteworthy, the conclusion is supported by the results. However, there is one major misconception regarding the ORAC assay and a couple of claims that are not supported by results/figures. In addition to those, I have several minor comments. See my concerns numbered below:

  1. Lines 239, 242, and 451: The authors should not refer to the ORAC as an “antilipoperoxidant” assay. This is not technically accurate. Although AAPH can trigger lipid peroxidation and it is used to do so in several different assays, the ORAC assay does not have a lipidic target. The target of the assay is a protein, fluorescein. Thus, it is incorrect to infer any antilipoperoxidant activity of tested sample unless a lipidic target is used. Please revise.
  2. Lines 260–261: This statement disagrees with the results shown in table 3. From the table, I would say the correct order is phacelia < acacia < sunflower < linden < chestnut. Please check.
  3. Lines 295–296: This statement is not supported by the results in Figure 2. The honey that is the poorest in total microelement is Amorpha. Please check.
  4. Line 19: It would be better to refer to “absorbance” or “darkness”, instead of “color”. Saying that color correlated positively with another variable is not informative.
  5. Line 362: Strictly speaking, the DPPH assay cannot contribute to the color of the honey in any manner. Did you mean that the color is a good predictor of the results of the DPPH assay? Please check.
  6. Line 24: Here in the abstract, it is better to name each parameter (or the most relevant) you are referring to, instead of saying “all other parameters”. The reason is that in the abstract (which should stand alone) the reader is not aware of all measurements that were made in the study.
  7. Lines 84-85: This sentence should be re-phrased for accuracy. Only 8 honeys were analyzed in triplicate. Saying that 24 samples were analyzed is misleading.
  8. Line 154: Here again, it is better to refer to “absorbance” than “color”.
  9. Lines 208–210: Make sure this statement is correct, since it might trigger negative responses from the authors.
  10. Lines 266–269: This sentence is confusing. I could not understand what the authors meant. Please revise.
  11. Figure 2: For clarity, consider drawing thin or dashed lines to denote the origin (zero) of both X-axis and Y-axis.
  12. Lines 429–432: Please add at which wavelength the absorbance was measured in the text.
  13. Lines 438–450: What was the Trolox used for here in the DPPH assay if results are simply expressed as IC50 values?

Author Response

Response to the comments from Reviewer 2

Thank you for the thorough review of the manuscript. Please find below the detailed answers to all the concerns raised.

The manuscript by Kocsis and colleagues characterized  eight selected Hungarian honeys based on their botanical origin, mineral content and the in vitro radical scavenging activity. The motivation behind the study is to generate a comprehensive picture of the characteristics of honeys produced in Hungary, one of the major honey producers in Europe. As expected from previous studies, the darkness of honey samples generally correlated with many other measurements, such as the results from the TRC, DPPH, and ORAC assays, as well as the content of many minerals (e.g., K, Ca, P, S, and Mg). The introduction is well-written and covers key literature in the field. The methodology is technically sound and the results are logically presented and explained. Despite some degree of speculation, the statements in the discussion and conclusions are supported by the experimental data and literature. Although it would be more interesting to also have performed antioxidant measurements using cell-based assays, the generated dataset tells a complete story and the manuscript reached the objective it was set out to achieve. Noteworthy, the conclusion is supported by the results. However, there is one major misconception regarding the ORAC assay and a couple of claims that are not supported by results/figures. In addition to those, I have several minor comments. See my concerns numbered below:

Many thanks for your positive comments and for calling our attention to improve the manuscript with some corrections and our honey research completed with cell-based assays.

There are publications in the literature (e.g. Sarfraz Ahmed et al. "Honey as a Potential Natural Antioxidant Medicine: An Insight into Its Molecular Mechanisms of Action", Oxidative Medicine and Cellular Longevity, vol. 2018, Article ID 8367846, 19 pages, 2018. https://doi.org/10.1155/2018/8367846) regarding honey as a potent antitumor, antidiabetic, anti-inflammatory agent with strong wound healing acceleration activity as well. Honey is also considered to possess antifungal, antibacterial, antiviral and even anti-biofilm activities.

In our laboratory, we have the facility for the measurement of both mammalian and/or microbial cell-based assays (DCFH and dihydrorhodamine 123 intracellular fluorescence, chemiluminescent H2O2 determination, flow cytometric apoptosis/necrosis detection, etc.).

  1. Lines 239, 242, and 451: The authors should not refer to the ORAC as an “antilipoperoxidant” assay. This is not technically accurate. Although AAPH can trigger lipid peroxidation and it is used to do so in several different assays, the ORAC assay does not have a lipidic target. The target of the assay is a protein, fluorescein. Thus, it is incorrect to infer any antilipoperoxidant activity of tested sample unless a lipidic target is used. Please revise.

Response: Thank you for your suggestion. We agree and deleted the misleading expression in the text.

  1. Lines 260–261: This statement disagrees with the results shown in table 3. From the table, I would say the correct order is phacelia < acacia < sunflower < linden < chestnut. Please check.

Response: Thank you for your correction, the right order is as you wrote above. The text was changed accordingly.

  1. Lines 295–296: This statement is not supported by the results in Figure 2. The honey that is the poorest in total microelement is Amorpha. Please check.

Response: You are right again, many thanks, we corrected the name and corresponding data of this honey sample.

  1. Line 19: It would be better to refer to “absorbance” or “darkness”, instead of “color”. Saying that color correlated positively with another variable is not informative.

Response: We agree and changed the term color into absorbance.

  1. Line 362: Strictly speaking, the DPPH assay cannot contribute to the color of the honey in any manner. Did you mean that the color is a good predictor of the results of the DPPH assay? Please check.

Response: Thank you for your comment. The text was rephrased as follows: “The DPPH assay was the strongest predicting factor regarding the absorbance (color) of the honey samples…”

  1. Line 24: Here in the abstract, it is better to name each parameter (or the most relevant) you are referring to, instead of saying “all other parameters”. The reason is that in the abstract (which should stand alone) the reader is not aware of all measurements that were made in the study.

Response: Thanks for this comment, we rephrased this sentence as follows: “The black meadow sage honey performed best regarding content of other elements and antioxidant activity.”

  1. Lines 84-85: This sentence should be re-phrased for accuracy. Only 8 honeys were analyzed in triplicate. Saying that 24 samples were analyzed is misleading.

Response: Thank you, your comment is true. The total number of replicates was 24 therefore we deleted 24 from the text.

  1. Line 154: Here again, it is better to refer to “absorbance” than “color”.

Response: We agree and changed the word in the text.

  1. Lines 208–210: Make sure this statement is correct, since it might trigger negative responses from the authors.

Response: Thank you for your goodwill warning, the sentence has been deleted.

  1. Lines 266–269: This sentence is confusing. I could not understand what the authors meant. Please revise.

Response: The sentence was rephrased as follows: “The absolute values obtained for the above mentioned five honey types were not uniform among the studied Hungarian honeys [23,51] however, the tendency regarding their average amounts of Ca and P showed considerable correlation.

  1. Figure 2: For clarity, consider drawing thin or dashed lines to denote the origin (zero) of both X-axis and Y-axis.

Response: Thank you, the figure was redrawn as you requested.

  1. Lines 429–432: Please add at which wavelength the absorbance was measured in the text.

Response: Thank you for your suggestion, the sentence has been completed with the wavelength parameters.

  1. Lines 438–450: What was the Trolox used for here in the DPPH assay if results are simply expressed as IC50 values?

Response: Trolox was used only for checking the linearity of the assay. However, we calculated the IC50 for Trolox as well: 25.83±1.86 µg/ml-1. As it can be seen from the comparison of the IC50s between Trolox and honey samples, the honeys showed less antioxidant capacities than the pure Trolox standard. This was the reason why the Trolox value was not included into the table.

This manuscript is a resubmission of an earlier submission. The following is a list of the peer review reports and author responses from that submission.